# Periodical Changes of Feces Microbiota and Its Relationship with Nutrient Digestibility in Early Lambs

**DOI:** 10.3390/ani12141770

**Published:** 2022-07-11

**Authors:** Yongliang Huang, Guoxiu Wang, Chong Li, Weimin Wang, Xiaoxue Zhang, Xiaojuan Wang, Deyin Zhang, Zhanyu Chen, Panpan Cui, Zongwu Ma

**Affiliations:** College of Animal Science and Technology, Gansu Agricultural University, Lanzhou 730070, China; huangyl_gau@163.com (Y.H.); wanggx@gsau.edu.cn (G.W.); wangwm@gsau.edu.cn (W.W.); zhangxx@gsau.edu.cn (X.Z.); wangxj@gsau.edu.cn (X.W.); zdy1213@163.com (D.Z.); chenzy_gsau@163.com (Z.C.); cpp0314@163.com (P.C.); gsaumzw980406@163.com (Z.M.)

**Keywords:** lambs, fecal microorganism, microbial diversity, digestibility

## Abstract

**Simple Summary:**

The birth-to-weaning period is critical for the development of digestibility and the intestinal microflora of lambs. This study aimed to analyze the developmental changes of the intestinal microflora and the host-apparent digestibility in this critical stage, and the interaction and causal relationship between them. The results showed that the fecal microbial diversity, and the abundance of some bacteria showed regular changes before lambs were 49 days old. Rapid changes in nutrient intake and digestibility are major factors that influence the fecal microbiota by affecting the composition of fermentable substrates in feces. Moreover, some bacteria were not affected by the fecal nutrient content, which is an important environmental factor, but were closely related to the lambs’ nutrient-apparent digestibility. These bacteria might have a regulatory role in nutrient digestibility.

**Abstract:**

Early mammal gut microorganism colonization affects host health, the feed conversion rate, and production performance. Herein, we explored how fecal microbiota develops and the interactions between microorganisms and nutrients. The lambs were separated from ewes at 7 days old, artificial feeding with milk replacer completely replaced lactation, and the starter diet was added. At 21 days old, the lambs were fed with complete starter and milk replacer was stopped. At day 7, 21, 35, and 49 after birth, fecal samples were collected. Then 16S rRNA gene sequencing in the fecal samples revealed that the alpha diversity increased significantly with age. Principal coordinates analysis showed clear clustering by age (*p <* 0.05). At the genus level, the relative abundance of 8 genera declined, 12 genera increased (*p <* 0.1), and 4 genera changed dramatically with age (*p <* 0.05). The apparent digestibility of dry matter, protein, fat, neutral detergent fiber, and acid detergent fiber significantly decreased by 21.77%, 23.15%, 23.62%, 19.38%, and 45.24%, respectively, from 7 to 35 days of age (*p* < 0.05), but not thereafter (*p* > 0.05). Fecal nutrient contents affected the abundance of bacterial genera (*p* < 0.05). *Enterobacteriaceae_unclassified*, *Clostridium XlVb*, *Bifidobacterium*, and other genera had no relationship with the fecal nutrient content; however, they were closely related to nutrient intake and digestibility, possibly promoting nutrient digestion. Our results suggested that nutrient digestion of young lambs changed rapidly, which was closely related to intestinal microbial colonization.

## 1. Introduction

Early gut colonization and succession of microorganisms play a significant role in animal health, the feed conversion rate, and feed intake, and is of great significance for animals to grow and develop during their whole life. In human and mouse model studies, the host is dependent on the distal gut microbiome to provide the capacity for carbohydrate structure digestion and absorption, the modulation of bile acid conjugative patterns, fat emulsification and absorption, and the modulation of lipid metabolism. However, the nutrient and energy value of food is not absolute, but is affected partly by the digestive capacity of an individual’s microbiota [1,2,3]. Nevertheless, early studies of the rumen microbial communities focused on the types and function of microorganisms that affect the host’s production performance, blood physiological parameters, and rumen development, using both culture and genetic approaches; however, we found that relatively few studies have been published regarding intestinal tract microorganism [4]. Therefore, studying the relationship between gut microbes with nutrient digestibility is critical to improve production performance, aiming to reduce farming costs and increase benefits to lambs.

The digestive system of ruminants is inhabited by many species and types of microorganisms, and their main role is decomposition of nutrients [5]. In the early part of life, both rumen and intestinal microbes undergo rapid changes in colonization, succession, and stabilization. The dominance of strictly anaerobic species very soon after birth (2 d) and the early appearance (first week) of populations of cellulolytic and methanogenic bacteria, show that the rumen ecosystem is quickly established after birth before the rumen itself becomes functional [6]. The microbial community and the function of rumen are not well-established before 20 days of age; therefore, there is a degree of plasticity in the rumen bacterial community during the first 20 days of post-natal development in lambs, and this might provide an opportunity for interventions to improve rumen fermentation [7]. Significantly, Furman et al. observed that members of the core successional microbiome appeared earlier than all others, within the first 140 days of life, with most of them being introduced during the first days after birth in cattle [8]. The intestinal microbiota is a complex community of microorganisms that live in the intestinal tract, playing an important role in barrier function and providing many benefits for the host, including defending against pathogens, using all available nutrients, and secreting compounds that kill or inhibit unwelcome organisms that would compete for nutrients [9]. Studies have shown that homeostasis of gastrointestinal micro-organisms is threatened by many external factors, the most important of which are heat stress, psychological stress, environment, and diet [10]. Furthermore, different weaning strategies have different effects on the gastrointestinal microflora of early lambs [11].

The development of the nutrient digestibility of lambs is very important for their whole life. The early postnatal and weaning stages are key for lamb digestive tract development [12]. However, the interaction and causal links between the gastrointestinal microflora and the digestibility of the host are still uncertain. On the one hand, the intake of all nutrients increases with age, and with the increase of solid feed intake, intestinal microbial diversity index and bacterial abundance change significantly before 49 days of age [13]. On the other hand, gastrointestinal microbes play an important role in the digestion and absorption of nutrients. Research has shown that a higher abundance of functional bacteria in the rumen improves forage digestibility, while producing high concentrations of NH_3_-N and short volatile fatty acids (VFAs) to rapidly improve growth performance [14]. For example, Prevotella bacteria appear to be associated with propionic acid production and play a pivotal role in degrading and utilizing plant non-cellulosic polysaccharides, protein, starch, and xylans [15]. Bacteria such as Verrucomicrobia contain a wide range of glycoside hydrolases, which play an important role in the decomposition of polysaccharides and cellobiose [16]. Researchers have been mainly focused on understanding the rumen microbiota’s contribution to the host in the last decade. However, each region of the gastrointestinal is spatially specialized depending on factors including physiology substrate availabilities, retention time of digesta, and pH levels [17]. These factors are all expected to have a profound impact on the local microbial assemblages and functions, thereby affecting the digestive, immunological, metabolic, and endocrinological processes in ruminants [18]. The rectal fecal flora of ruminants is highly similar to that of the colon and cecum [19], and fecal samples are easy to collect. Increased understanding of the relationship between fecal microorganisms and nutrient utilization in ruminants has a positive effect on reducing feed cost and improving animal productivity. However, it remains to be further explored whether the colonization of the intestinal flora of young lambs has an important effect on the development of nutrient digestion and absorption.

Thus, the objective of this study was to evaluate the periodical changes of growth, nutrient intake, digestibility, fecal nutrient content, and fecal microbiota, and to explore the relationship between microbial diversity and nutrient digestibility in early lambs. We hypothesized that with the increase of starter intake, nutrient digestibility changes rapidly, which alters the composition of fermentation substrates and thus affects the fecal microbiota.

## 2. Materials and Methods

### 2.1. Experiment Design and Animal Management

According to the birth weight principle (mean ± SD: 3.29 ± 0.68 kg), six healthy neonatal male Hu lambs were selected from a commercial sheep farm (Jinchang Zhongtian Sheep Industry Co. Ltd., Jinchang, China). From birth to 3 days, the lambs were kept indoors with ewes to ensure adequate colostrum intake. The lambs were trained to use the nipple bottle containing reconstructed milk replacer (crude protein 23.22%, fat 13.20%, air-dried basis; Feed Research Institute, Chinese Academy of Agricultural Sciences, Beijing, China; Table 1) from 4 to 6 days. Milk replacer was reconstituted at 200 g/L in water and provided for lambs at a temperature of 40 °C. Each lamb was fed 50 mL of milk replacer three times a day (at 09:00, 15:00, and 21:00). At 7 days old, the lambs were separated from ewes and placed in individual pens (0.65 m × 1.10 m), and artificial feeding with milk replacer (2% of average body weight per day) completely replaced lactation, and the starter diet was added. At 21 days old, the lambs were fed with complete starter and milk replacer was stopped. These diets, published in China, are formulated to meet the requirements of the feeding standard of meat-producing sheep and goats (NYT816-2004), and their formula and nutritional composition are shown in Table 1. All lambs had free access to the starter diet and clean water. All lambs were weighed every 14 days to calculate average body weight and average daily gain. The starter intake of each lamb was recorded daily as the difference between offered and refused feed and the average intake was calculated.

### 2.2. Sample Collection and Measurement of Nutrient Digestion

Fecal samples for total microbial DNA extraction were obtained by rectal stimulation before morning feeding from lambs at 7, 21, 35, and 49 days old, and were stored in sterile test tubes at −80 °C. Apparent total tract digestibility was determined by total fecal collection method [20]. At 7–10, 18–21, 32–35, and 46–49 days old, the starter intake of each lamb was recorded daily as the difference between offered and refused feed, and all feces were collected and separated from urine with a slatted floor and gauze. The feces were weighed daily and, pooled for each 3-d period. A proportion of the feces was divided, and stored in 10% sulfuric acid for crude protein (CP) determination. Other fecal samples were dried at 65 °C to calculate the initial moisture and stored at room temperature for dry matter (DM), crude fat ether extract (EE), neutral detergent fiber (NDF), acid detergent fiber (ADF), and starch determination. The milk replacer, starter and feces were analyzed for DM(drying at 105 °C), CP(AOAC International, 2000), EE(AOAC International, 2000) [21], NDF and ADF following a previously described method with heat-stable alpha-amylase and sodium sulfate used in the NDF procedure [22], and starch using a commercial assay kit (Solarbio, Shanghai, China) according to the manufacturer’s instructions. The apparent digestibility of protein, starch, fat, DM, ADF, and NDF was calculated in the following Equation:AD = [(Fi − Ff)/Fi] × 100%,
where AD is the apparent digestibility of protein, starch, fat, DM, ADF, or NDF (%), Fi is the intake of protein, starch, fat, DM, ADF or NDF (g), and Ff is the fecal output of protein, starch, fat, DM, ADF or NDF (g).

### 2.3. Bacterial DNA Extraction

DNA from different samples was extracted by using an E.Z.N.A. ^®^Stool DNA Kit (D4015, Omega Bio-tek, Norcross, GA, USA) according to manufacturer’s instructions. The kit contains a reagent designed to recover DNA from trace amounts of sample and has been shown to be effective for the preparation of DNA of most bacteria. Nuclear-free water was used as the blank control. The total DNA was eluted in 50 µL of Elution buffer and stored at −80 °C until measurement by using PCR by LC-Bio Technology Co., Ltd., HangZhou, China.

### 2.4. PCR Amplification and 16S rDNA Sequencing

The V4 region of the prokaryotic (bacterial and archaeal) small-subunit (16S) rRNA gene was amplified with primers 515F (5′-GTGYCAGCMGCCGCGGTAA-3′) and 806R (5′-GGACTACHVGGGTWTCTAAT-3′). The 5′ ends of the primers were tagged with specific barcodes per sample and with universal sequencing primers.

PCR amplification was performed in a total volume of 25 µL. The reaction mixture contained 25 ng of template DNA, 12.5 µL of PCR Premix, 2.5 µL of each primer, and PCR-grade water to adjust the volume. The PCR conditions to amplify the prokaryotic 16S fragments consisted of an initial denaturation at 98 °C for 30 s; 35 cycles of denaturation at 98 °C for 10 s, annealing at 54 °C/52 °C for 30 s, and extension at 72 °C for 45 s; and a final extension at 72 °C for 10 min. The PCR products were confirmed by using 2% agarose gel electrophoresis. Throughout the DNA extraction process, ultrapure water, instead of a sample solution, was used to exclude the possibility of false-positive PCR results, as a negative control. The PCR products were purified by using AMPure XT beads (Beckman Coulter Genomics, Danvers, MA, USA) and quantified by using a Qubit instrument (Invitrogen, Waltham, MA, USA). Amplicon pools were prepared for sequencing and the size and quantity of the amplicon library were assessed on an Agilent 2100 Bioanalyzer (Agilent, Santa Clara, CA, USA) and with a Library Quantification Kit for Illumina (Kapa Biosciences, Woburn, MA, USA), respectively. A PhiX Control library (v3) (Illumina Inc., San Diego, CA, USA) was combined with the amplicon library. The libraries were sequenced either as 250-bp paired-end reads (250PE) MiSeq runs, and one library was sequenced with both protocols by using the standard Illumina sequencing primers, eliminating the need for a third (or fourth) index read. Samples were sequenced on the Illumina MiSeq platform according to the manufacturer’s recommendations, provided by LC-Bio. The sequencing data were deposited into the Sequence Read Archive (SRA) of NCBI and can be accessed via accession number PRJNA836702.

### 2.5. Sequence and Statistical Analysis

Paired-end reads were assigned to samples based on their unique barcodes and truncated by cutting off the barcode and primer sequence. Paired-end reads were merged by using FLASH (32.0.0.371). Quality filtering on the raw tags were performed under specific filtering conditions to obtain high-quality clean tags, according to FastQC (V 0.10.1). Chimeric sequences were filtered by using Verseach software (v2.3.4). Sequences with ≥97% similarity were assigned to the same operational taxonomic units (OTUs) by using Verseach (v2.3.4). Representative sequences were chosen for each OTU, and taxonomic data were then assigned to each representative sequence by using the RDP (Ribosomal Database Project) classifier. The differences in the dominant species in the different groups were identified and multiple sequence alignment was conducted by using the PyNAST software to study the phylogenetic relationship of the different OTUs. OTU abundance information was normalized by using the number of sequences in the sample with the least number of sequences as a standard. Alpha diversity analysis was applied to assess the complexity of species diversity for a sample by using four indices: Chao1, Shannon, Simpson, and Observed species. These indices were calculated by using QIIME (Version 1.8.0). Beta diversity analysis was used to evaluate differences in species complexity in the samples. Beta diversity was calculated by principal coordinates analysis (PCoA) and cluster analysis by QIIME software (Version 1.8.0).

The data of microbial diversity indices (among these diversity indices, the Shannon index measures uncertainty about the identity of species in the sample, and its units quantify information, while the Simpson measures a probability, specifically, the probability that two individuals, drawn randomly from the sample, will be of different species [23]. Coverage is the proportion of individuals belonging to undiscovered species in a community that can be reliably estimated based solely on the frequency of species already present in the sample [24]. Chao1 was asymptotic richness estimators and could predict the community diversity [25]), bacterial densities, growth performance, apparent digestibility, daily intake, apparent digestion, daily excretion, and fecal nutrient contents were analyzed by using one-way ANOVA and the least significant difference (LSD) post hoc tests in SPSS software (version 25.0; IBM Corp., Armonk, NY, USA) with 5 degrees of freedom. The following statistical model was used: *Y_ij_ = μ + Ai + e_ij_*, where *Y* is the microbial diversity indices, bacterial densities, growth performance, apparent digestibility, daily intake, apparent digestion, daily excretion or fecal nutrient contents; *μ* is the mean; *A* is the age; and *e* is the residual error. Spearman correlation coefficients were used to evaluate relationships among the most abundant genera and nutrient contents in feces and nutrient digestion using the R software (version 4.1.1). Statistical significance was set at *p* < 0.05, *p* < 0.001 indicated an extremely significant difference.

## 3. Results

### 3.1. Microbial Diversity Analysis

#### 3.1.1. Alpha Diversity

To investigate the periodic changes in the fecal microbiota in early lambs, this study used 16S rRNA gene sequencing of feces samples to compare differences of intestinal microbiota between 7 days (D7), 21 days (D21), 35 days (D35) and 49 days (D49). An average of 50,747 V3–V4 16SrRNA gene sequence reads were obtained for each sample from the early lambs after quality control. Rarefaction curves demonstrated that almost all the microbes were detected in feces of early lambs (Figure 1a). The overall number of OTUs was 1628, and 372 shared OTUs could be detected in all groups. There were 30, 42, 25, and 43 endemic species in the D7, D21, D35, and D49 samples (Figure 1b). The Observed species and Chao1 indices increased with age; however, the Shannon index increased from 7 to 21 days of age and then start to decrease after weaning (Table 2).

#### 3.1.2. Comparison between Microbial Communities (Beta Diversity)

The unweighted UniFrac distances were used to derive principal coordinates (PCoA) (Figure 2a). Based on PCoA, the unweighted measures showed clear clustering by age, and microbial communities appeared to become more similar between individual early lambs at later sample times (D35 and D49), indicating a convergence of the microbial populations from individual lambs over time. However, the weighted UniFrac analysis (Figure 2b) found no difference in microbial community dispersion in individuals between sample time points.

#### 3.1.3. Phylogenetic Composition of Fecal Microorganism Communities

When the phylum-level results for each of the six individual lambs were averaged together by sample time point (Figure 3), Bacteroidetes was the dominant phylum at each of the four timepoints (ranging from 29.22% relative abundance at D7 to 47.11% at D49), followed by Firmicutes (50.20% at D7 to 27.68% at D35), Proteobacteria (appeared at D7 (9.94%), ranging from 11.66% at D21 to 5.79% at D35, then, increasing to 6.09% at D49). Verrucomicrobia decreased from 10.07% at D7 to 5.07% D21, and then decreased from 12.98% at D35 to 5.78% at D49 (*p* < 0.05). However, the relative abundance of Actinobacteria, Euryarchaeota, and Bacteria_unclassified increased gradually from D7 to D49. Furthermore, Fusobacteria and Synergistetes first appeared at D21, and then remained until the end of the experiment. Fusobacteria, Elusimicrobia, Candidatus_Saccharibacteria, Cyanobacteria, and Candidatus_Melainabacteria first appeared at D35. Finally, Candidatus Saccharibacteria, and Tenericutes first appeared at D49.

Table 3 shows the genus-level top 20 taxonomic composition of the gut microbial communities. The relative abundance of *Porphyromonadaceae*_unclassified (*p* = 0.018) (ranging from 0.3417% at D7 to 22.0550% at D35) and *Clostridium XlVb* (*p* = 0.030, ranging from 0.12% at D7 to 3.87% at D21) increased significantly with age. By contrast, *Lachnospiraceae*_unclassified (*p =* 0.034, ranging from 19.6933% at D7 to 5.1425% at D47) and *Clostridium XlVa* (*p =* 0.009, ranging from 5.2267% at D7 to 0.4125% at D35) decreased significantly with age, especially between D21 and D35. However, other genera showed no significant difference among the different sample times (*p >* 0.05). A heat map of the genus-level taxonomic composition indicated that the relatively high or low abundance genera were significantly different among the different groups (Figure 4).

### 3.2. Dynamic Changes of Nutrient Digestibility in Lambs and Its Relationship with Fecal Microorganisms

The body weight, average daily gain and intake of lambs are presented in Table 4. The starter intake and body weight increased significantly with age (both *p* < 0.001), and average daily gain decreased from 7 to 21 days of age, and then gradually increased (*p* < 0.001). Based on the effects of different sampling times on apparent digestibility in early lambs (Table 5), we observed that the digestibility of protein decreased significantly with age (*p* < 0.001); however, the daily intake, daily digestion and daily excretion of protein increased significantly with age (all *p* < 0.001). Starch digestibility showed no significant difference among the different sample times (*p* = 0.329); however, its daily intake, daily digestion and daily excretion increased significantly after 21 days (all *p* < 0.001), daily intake, daily digestion and daily excretion of fat increased significantly with age (all *p* < 0.001); however, the daily intake and daily digestion of fat had bidirectional dynamics (both *p* < 0.001). The digestibility, daily intake, and daily excretion of DM significantly increased with age (all *p* < 0.001), whereas digestibility decreased significantly (*p* < 0.001) and showed bidirectional dynamics. The digestibility of NDF decreased significantly (*p* = 0.005) and showed bidirectional dynamics with age; however, its daily intake, digestibility and daily excretion increased significantly with age (all *p* < 0.001). The digestibility of ADF decreased significantly with age (*p* < 0.001), particularly at D7 and D21, whereas its daily intake, daily digestion and daily excretion increased significantly with age (all *p* < 0.001).

Correlation analysis of the relative abundances of the top 20 genus-level taxonomic composition and apparent digestibility was performed (Figure 5). The result showed that many genera correlated significantly with nutrient digestion (*p* < 0.05), and those related to intake and digestion were clearly different from those related to the digestibility of DM, fat, protein, ADF and NDF, and were clustered together. Some genera showed positive correlations with ADF, NDF, DM, starch daily intake and apparent digestion, including *Porphyromonadaceae*_unclassified, *Clostridiales*_unclassified and *Bifidobacterium* (*p* < 0.05). Conversely, other genera correlated inversely with the same indices, including *Lachnospiraceae*_unclassified and *Clostridium XlVa* (*p* < 0.05). Besides, ADF and NDF apparent digestibility correlated positively with *Enterobacteriaceae*_unclassified, but negatively with *Ruminococcaceae*_unclassified and *Methanobrevibacter* (*p* < 0.05). Fat daily intake and apparent digestion correlated positively and significantly with *Clostridium XlVb*, but negatively with *Methanobrevibacter* (*p* < 0.05).

The fecal fat content decreased significantly with age at E7 and E21 (*p =* 0.016). By contrast, the fecal DM, protein, ADF, and NDF contents all increased significantly with age, especially between the 21- and 35-day samples (all *p* < 0.001). However, the fecal starch content showed no significant difference among the different sample times (*p* = 0.857) (Table 6).

Correlation analysis of the relative abundances of the top 20 genus-level taxonomic composition and fecal nutrient contents is shown in Figure 6. The relative abundance of *Parabacteroides* correlated significantly and positively with the ADF (*p* = 0.0325) and NDF (*p* = 0.0252) contents, but negatively with the fat content (*p* = 0.0487). The relative abundance of *Olsenella* correlated significantly and positively with the ADF (*p* = 0.0106) and NDF (*p* = 0.0061) contents, but negatively with the protein content (*p* = 0.0318). The abundances of *Methanobrevibacter* and *Porphyromonadaceae*_unclassified correlated positively with the ADF (*p* = 0.0344, *p* = 0.0148) and NDF (*p* = 0.0427, *p* = 0.0245) contents, but negatively with the protein content (*p* = 0.0318, *p* = 0.0190). The abundance of *Clostridiales*_unclassified correlated positively with the ADF (*p* = 0.0470) and DM (*p* = 0.0025) contents, but negatively with the protein content (*p* = 0.0045). The abundance of *Blautia* correlated positively with the DM (*p* = 0.0495), fat (*p* = 0.0017), and protein (*p* = 0.0002) contents. The abundance of *Lachnospiraceae*_unclassified correlated negatively with the ADF content (*p* = 0.0214). The abundance of *Clostridium.XlVa* correlated negatively with the ADF content (*p* = 0.0288), and positively with the DM content (*p* = 0.0234). The abundance of Lactobacillus correlated positively with the fat content (*p* = 0.0069).

## 4. Discussion

Fecal microbes can represent the intestinal flora to study the relationship between the periodic changes of intestinal flora and digestibility in early lambs. Previous research has shown that that fecal microbiota is highly variable in the early life of calves [26], playing an important role in body health [27], digestive processes, and immune response, and is affected by many factors, such as age [28], diet [28], the environment [29], and weaning [11]. Furthermore, a study in house mice observed that 93.3% of OTUs were shared between fecal and lower gastrointestinal samples [30]. Another study indicated that gastrointestinal origin is a primary determinant for the fecal microbiota composition [31]. These results indicated that fecal samples have good potential to identify microbial members derived from the digestive tract. Therefore, a more detailed understanding of the progression of the early lamb fecal microbiome from the neonatal period, including weaning and the commencement of starter feed, will provide insights into what constitutes a stable microbiome at these crucial stages of growth and development.

The diversity of fecal flora showed an increasing trend until the end of the experiment. In this study, alpha diversity indices (Observed species, Shannon, Simpson and Chao1) of the fecal microbiota increased with age until the end of the experiment, which corresponded with the process of microbial colonization and development observed in the gut of ruminants [32]. A highly diverse gut microbiota is generally considered beneficial for host health and is regarded as a sign of a mature gut microbiota [33]. However, we observed a decreased alpha diversity during the early lambs after weaning (D21), the main reason for which might be sudden diet transition from milk replacer to solid feed after weaning [34]. Therefore, the development of the intestinal microflora of lambs up to 49 days old is a key stage, in which lambs drink less milk (highly digestible) and more starter (less digestible) over time, and microbes updated to change in nutrients available in the intestinal tract.

Transformation to solid feed changes the fecal microflora. Consistent with previous studies, this study demonstrated that Bacteroidetes and Firmicutes were the two most dominant phyla in the fecal microbial communities of early lambs. A study based on human infants indicated that Bacteroidetes and Firmicutes were the most prevalent phyla [35]. Our results revealed decreases in the relative abundances of five phyla (Firmicutes, Verrucomicrobia, Proteobacteria, and Synergistetes) and increases in the relative abundances of four phyla (Bacteroidetes, Actinobacteria, Euryarchaeota and Fusobacteria) with the age of early lambs. These microorganisms changed mainly because of introduction of fiber in the solid feed. This result might be related to the gradual maturation and stabilization of the intestinal flora. At the genus level, the relative abundances of 8 genera declined, whereas the relative abundances of 12 microbe genera increased with increasing age and weaning of the early lambs. The relative abundances of only four genera changed dramatically: *Lachnospiraceae* and *Clostridium XlVa* numbers decreased, whereas *Akkermansia* and *Clostridium XlVb* numbers increased significantly. Reports suggested that the presence of *Clostridium XlVa* might be the main cause of diarrhea in early lambs [36,37]. Furthermore, proteolysis is common among *Clostridia* species [38]; therefore, the changes in the levels of these species might have been be caused by the decrease in the protein composition in milk after weaning and increased body resistance with the growth of lambs. In addition, *Lachnospiraceae* ferment diverse plant polysaccharides to short-chain fatty acids and alcohols [39], thus the decline in abundance of *Lachnospiraceae*_unclassified might be related to the gradual increase in fibrous feed intake. Indeed, the results showed that the abundance of Lachnospiraceae_unclassified correlated negatively with the fecal fiber content. Interestingly, the abundance of some bacteria decreased suddenly after weaning and then gradually increased toward D49, such as *lactobacillus* and *Bacteroidetes*. By contrast, *Akkermansia* suddenly increased after weaning and the gradually decreased toward D49. These results suggested that these genera are susceptible to the changes in diet composition. *Bifidobacterium* and *Lactobacillus* are representative probiotic bacteria. Both genera have been proven to beneficially affect intestinal health through different mechanisms and have anti-proliferative, proapoptotic, and anti-oxidant properties [40]. After a period of time, their abundance gradually recovered, indicating that the microflora adapted to change in diet composition after weaning.

With the increase in feed intake, the intake and daily digestion of DM, protein, starch, NDF, and ADF increased gradually; however, their digestibility decreased gradually. The apparent digestibility of DM, CP, starch, fat, NDF, and ADF significantly decreased by 21.77%, 23.15%, 23.62%, 19.38%, and 45.24%, respectively, from 7 to 35 days of age. This is consistent with the change rule of nutrient digestibility [12], which proposes that increased fiber results in a reduction of apparent nutrient digestibility [41,42]. Furthermore, we analyzed the correlation between nutrient digestibility and the fecal microflora, and the correlation between fecal nutrient contents and the fecal microflora, respectively, to explore the interaction and causality between host nutrient digestion and the microflora. Starch showed no statistical significance and correlation. Studies have reported that the main culprit is within the rumen, where more than 90% [43] of dietary plant cell walls and 20~90% of the starch are degraded [44], whereas nutrients entering the gut comprised recalcitrant carbohydrates. The abundances of *Parabacteroides*, *Olsenella*, *Methanobrevibacter*, *Porphyromonadaceae*_unclassified, and *Clostridiales*_unclassified were correlated significantly and negatively with fecal DM, protein, and fat contents, but positively with NDF and ADF contents, suggesting that the fecal fiber content affected the abundance of these bacteria, i.e., fiber comprised the fermentation substrate of these bacteria either directly or indirectly. The fiber content in feces is an important environmental factor for these bacteria. The niche of bacteria varies, and increased fermentable substrates supply may promote colonization and fermentation of specific bacteria. Although these bacteria also correlated significantly with nutrient digestibility, it might be that feed intake and digestibility affected the fecal nutrient content and thus affected the abundance of these bacteria. Moreover, we found that the abundance of some bacteria did not correlate with fecal nutrient content, which is an important environmental factor, but correlated significantly and positively with the digestibility of the corresponding nutrients, such as *Enterobacteriaceae*_*unclassified, Clostridium XlVb*, and *Bifidobacterium*. Although nutrient intake and digestion are major factors influencing the microbiota by influencing the types of substrates available in the digestive tract, some bacterial have the potential to interact directly with the host and affect nutrient digestion. However, the specific effects and mechanisms of these bacteria on nutrient digestion require further study.

## 5. Conclusions

The intestinal microflora of lambs changed significantly with age, and at up to 49 days old was still an important period of microflora development. The apparent digestibility of dry matter, protein, fat, neutral detergent fiber, and acid detergent fiber decreased rapidly with the increase of starter intake from 7 to 35 days of age, especially after weaning. Nutrient intake and digestion are major factors that influence the fecal microbiota by affecting the composition of fermentable substrates in feces. The findings expand our understanding of the gut symbiotic microbiota in ruminants and provide new insights for investigating the gut microbiota’s role in host production.

## Figures and Tables

**Figure 1 animals-12-01770-f001:**
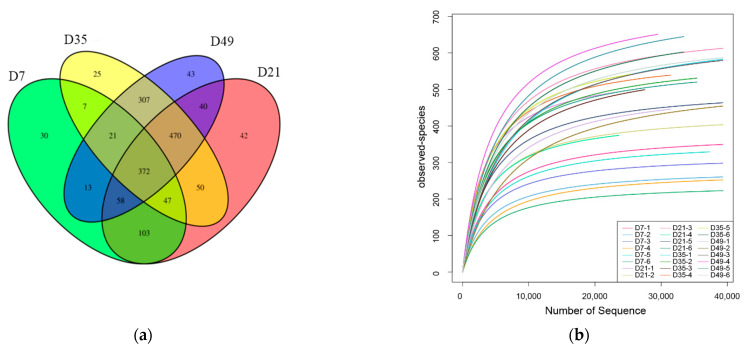
Microorganism rarefaction curves based on Observed (**a**) indices were used to assess the depth of coverage of each sample; each sample was distinguished by different colored lines. (**b**) Venn diagrams for Gut microbial operational taxonomic unit (OTU) compositions (D7, 7 days after birth; D21, 21 days after birth; D35, 35 days after birth; D49, 49 days after birth. At each time point, samples were obtained from six lambs).

**Figure 2 animals-12-01770-f002:**
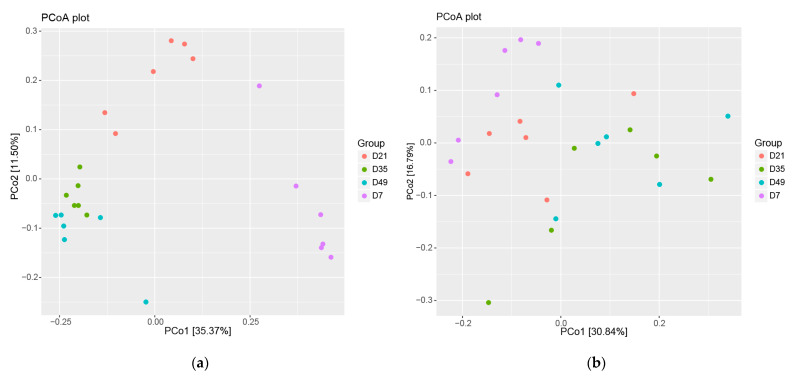
Development of gut microbial operational taxonomic units (OTUs) as the early lambs aged. (**a**) Unweighted UniFrac distances based on relative abundance of microbial OTUs, (**b**) Weighted UniFrac distances based on relative abundance of microbial OTUs.

**Figure 3 animals-12-01770-f003:**
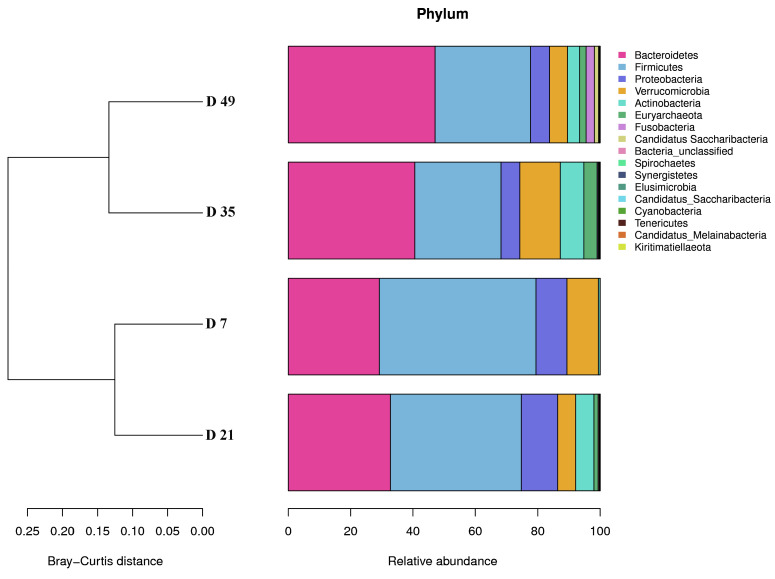
Relative abundance of the feal microbial compositions at the phylum level in early lambs. The mean value of relative abundance (given as a percentage) of the top 17 phyla at each timepoint are shown.

**Figure 4 animals-12-01770-f004:**
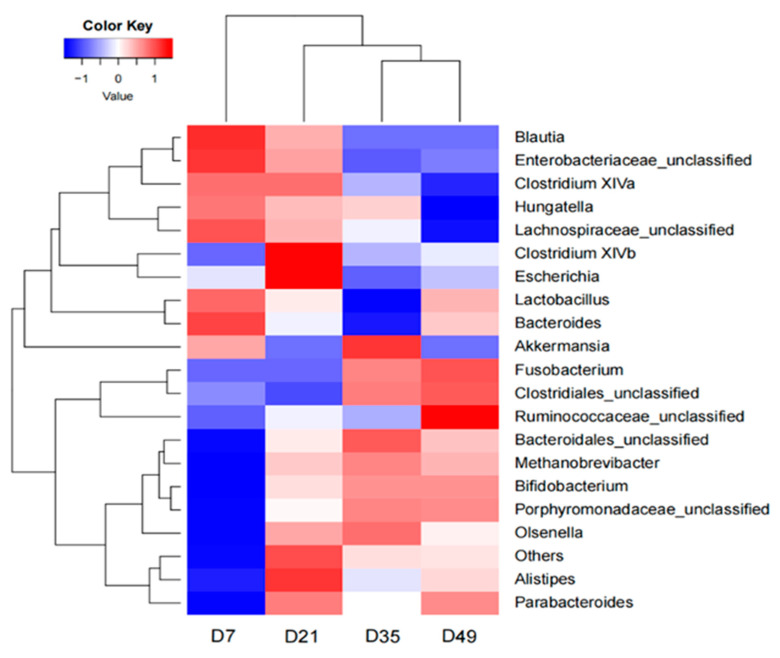
Heat map of the genus-level taxonomic composition in the early lambs at four sampling time points. The colors in the heat map represent the normalized relative abundances of genera (Log 10): red to blue represents more abundant to less abundant.

**Figure 5 animals-12-01770-f005:**
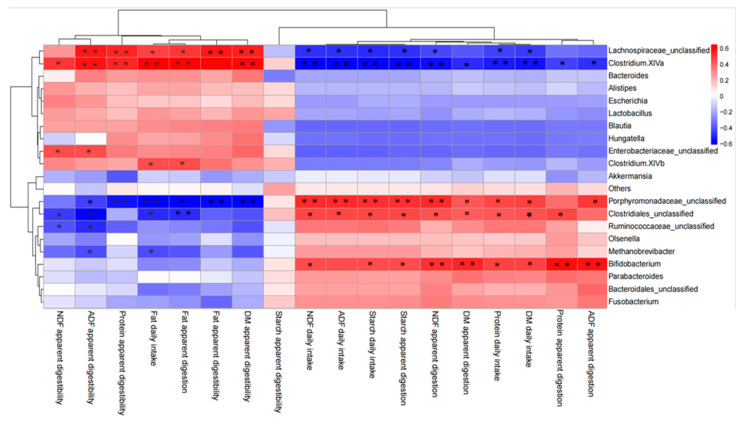
Correlation between the top 20 genus-level taxonomic composition and apparent digestibility. Data marked with * indicates a significant correlation (*p* < 0.05), and data marked with ** indicates an extremely significant difference (*p* < 0.01).

**Figure 6 animals-12-01770-f006:**
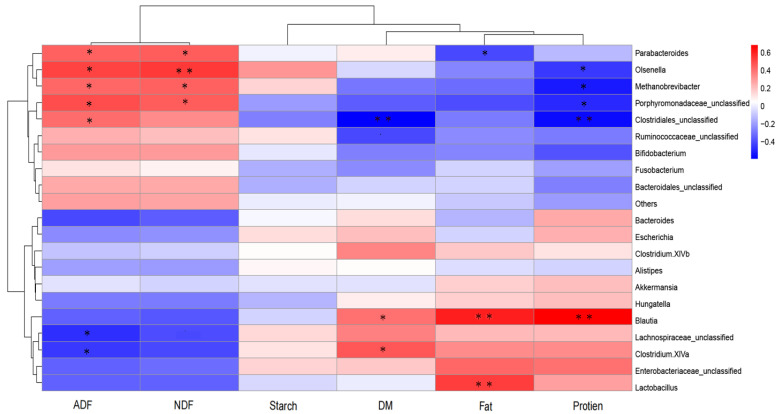
Correlation between the top 20 genus-level taxonomic composition and fecal nutrient contents. Data marked with * indicates a significant correlation (*p* < 0.05), and data marked with ** indicates an extremely significant difference (*p* < 0.01).

**Table 1 animals-12-01770-t001:** Ingredients and chemical composition of starter diet and milk replacer (air-dried basis).

Items	Starter ^1^	Milk Replacer
Ingredients [%]		
Alfalfa meal	18.50	
Corn	21.00	
Extruded corn	22.30	
Bran	6.00	
Soybean meal	21.50	
Extruded soybean	4.00	
Corn gluten meal	5.00	
Limestone	0.30	
Premix ^2^	1.00	
NaCl	0.40	
Total	100.00	
Chemical composition		
DM (%)	90.96	96.91
DE (MJ·kg^−1^)	13.01	/
CP (%)	19.50	23.22
Fat (%)	1.33	13.20
Starch (%)	33.10	0.00
NDF (%)	18.87	0.00
ADF (%)	8.60	0.00

Notes: ^1^ The starter was pelleted. ^2^ Premix provided the following per kg of the starter: 25 mg Fe as FeSO_4_·H_2_O; 40 mg Zn as ZnSO_4_·H_2_O; 8 mg Cu as CuSO_4_·5H_2_O; 40 mg Mn as MnSO_4_·H_2_O; 0.3 mg I as KI; 0.2 mg Se as Na_2_SeO_3_; 0.1 mg Co as CoCl_2_; 940 IU vitamin A;111 IU vitamin D;20 IU vitamin E, and; 0.02 mg vitamin B12.

**Table 2 animals-12-01770-t002:** Fecal microbial richness and diversity indices.

Diversity Indices	Groups				SME	*p*-Value
	D7	D21	D35	D49		
Observed index	267.67 ^a^	446.67 ^b^	525.17 ^b^	526.00 ^b^	25.445	<0.001
Shannon index	5.08 ^a^	6.18 ^b^	5.73 ^b^	5.72 ^ab^	0.135	0.026
Simpson index	0.94	0.96	0.93	0.95	0.008	0.376
Chao1 index	327.37 ^a^	579.17 ^b^	651.14 ^bc^	703.99 ^c^	33.452	<0.001

Notes: In the same row, values with the same or no letter superscripts indicate no significant difference (*p >* 0.05), while those with different small letter superscripts indicate a significant difference (*p <* 0.05).

**Table 3 animals-12-01770-t003:** Top 20 taxonomic composition at the genus-level of the gut microbial communities.

Genus	Groups				SEM	*p*-Value
	D7 (%)	D21 (%)	D35 (%)	D49 (%)		
*Bacteroides*	27.26	18.42	9.94	19.63	2.252	0.101
*Porphyromonadaceae*_unclassified	0.34 ^a^	5.47 ^a^	22.06 ^b^	19.06 ^b^	2.956	0.018
*Lachnospiraceae*_unclassified	19.69 ^b^	12.37 ^ab^	6.46 ^a^	5.14 ^a^	2.074	0.034
*Akkermansia*	10.07	5.79	16.24	5.94	2.722	0.067
*Ruminococcaceae*_unclassified	3.39	5.32	3.11	10.57	1.241	0.076
*Clostridiales*_unclassified	2.32	1.81	7.28	5.84	0.891	0.100
*Escherichia*	2.73	5.09	0.72	3.08	0.970	0.582
*Enterobacteriaceae*_unclassified	5.74	2.61	0.24	0.51	1.087	0.268
*Clostridium XlVa*	5.23 ^b^	5.13 ^b^	0.41 ^a^	0.56 ^a^	0.740	0.009
*Blautia*	7.23	2.32	0.29	0.25	1.322	0.204
*Alistipes*	0.26	4.63	0.76	1.51	0.650	0.074
*Olsenella*	0.11	2.41	3.64	2.15	0.586	0.266
*Methanobrevibacter*	0.01	1.31	4.03	2.46	0.621	0.192
*Lactobacillus*	3.27	0.46	0.01	0.85	0.457	0.056
*Parabacteroides*	0.59	1.72	0.65	1.80	0.268	0.211
*Bacteroidales*_*unclassified*	0.01	0.53	6.23	1.22	0.903	0.128
*Bifidobacterium*	0.01	0.66	2.45	1.90	0.516	0.384
*Clostridium XlVb*	0.12 ^a^	3.83 ^b^	0.21 ^a^	0.35 ^a^	0.460	0.030
*Fusobacterium*	0.01	0	0.13	2.18	0.613	0.460
*Hungatella*	2.73	0.26	0.16	0.01	0.471	0.115
Others	8.89	19.87	14.99	14.99	1.585	0.101

Notes: In the same row, values with the same or no letter superscripts indicate no significant difference (*p >* 0.05), while those with different small letter superscripts indicate a significant difference (*p <* 0.05).

**Table 4 animals-12-01770-t004:** Intake, body weight and average daily gain of lambs at different sample times.

Item	Groups				SEM	*p*-Value
	D7	D21	D35	D49		
Milk replacer intake (kg/d)	0.63	0.70	/	/	/	/
Starter intake (kg/d)	0.02 ^b^	0.06 ^b^	0.39 ^a^	0.49 ^a^	0.033	<0.001
Body weight (kg)	4.63 ^c^	5.72 ^c^	7.74 ^b^	10.17 ^a^	0.572	<0.001
Average daily gain (kg/d)	0.20 ^a^	0.08 ^b^	0.14 ^c^	0.17 ^cd^	0.024	<0.001

Notes: In the same row, values with the same or no letter superscripts indicate no significant difference (*p >* 0.05), while those with different small letter superscripts indicate a significant difference (*p <* 0.05).

**Table 5 animals-12-01770-t005:** Effects of different sampling times on apparent digestibility in early lambs.

Item		Groups				SEM	*p*-Value
		D7	D21	D35	D49		
Protein	Apparent digestibility (%)	86.53 ^b^	83.59 ^b^	66.50 ^a^	63.09 ^a^	2.630	<0.001
	Daily intake (g/d)	28.03 ^a^	39.40 ^a^	67.39 ^b^	94.84 ^c^	5.926	<0.001
	Apparent digestion (g/d)	24.26 ^a^	32.81 ^ab^	45.53 ^b^	59.81 ^c^	3.464	<0.001
	Daily excretion (g/d)	3.72 ^c^	6.58 ^c^	21.86 ^b^	35.04 ^a^	3.574	<0.001
Starch	Apparent digestibility (%)	84.99	88.65	86.95	87.33	0.695	0.329
	Daily intake (g/d)	5.01 ^a^	22.40 ^a^	114.37 ^b^	160.96 ^c^	14.038	<0.001
	Apparent digestion (g/d)	4.41 ^a^	19.91 ^a^	99.51 ^b^	140.82 ^c^	12.305	<0.001
	Daily excretion (g/d)	0.60 ^c^	2.49 ^c^	14.86 ^b^	20.14 ^a^	1.946	<0.001
Fat	Apparent digestibility (%)	91.09 ^b^	90.66 ^b^	69.57 ^a^	66.87 ^a^	2.558	<0.001
	Daily intake (g/d)	14.46 ^c^	16.70 ^d^	4.47 ^a^	6.49 ^b^	1.088	<0.001
	Apparent digestion (g/d)	13.17 ^c^	15.14 ^d^	3.13 ^a^	4.37 ^b^	1.108	<0.001
	Daily excretion (g/d)	1.29 ^b^	1.56 ^b^	1.34 ^b^	2.13 ^a^	0.186	<0.001
DM	Apparent digestibility (%)	89.40 ^b^	86.27 ^b^	69.94 ^a^	68.78 ^a^	2.044	<0.001
	Daily intake (g/d)	118.87 ^a^	171.67 ^a^	324.36 ^b^	456.50 ^c^	29.997	< 0.001
	Apparent digestion (g/d)	106.22 ^a^	147.58 ^a^	228.50 ^b^	315.14 ^c^	19.187	<0.001
	Daily excretion (g/d)	12.65 ^c^	24.09 ^c^	95.86 ^b^	141.35 ^a^	9.770	<0.001
NDF	Apparent digestibility (%)	67.44 ^b^	73.93 ^b^	54.37 ^a^	53.03 ^a^	2.702	0.005
	Daily intake (g/d)	2.87 ^a^	11.06 ^a^	65.44 ^b^	92.09 ^c^	8.113	<0.001
	Apparent digestion (g/d)	2.23 ^a^	7.97 ^a^	36.13 ^b^	49.32 ^c^	4.463	<0.001
	Daily excretion (g/d)	0.64 ^c^	3.09 ^c^	29.30 ^b^	42.77 ^a^	3.262	<0.001
ADF	Apparent digestibility (%)	69.45 ^b^	68.61 ^b^	38.03 ^a^	35.90 ^a^	4.165	<0.001
	Daily intake (g/d)	1.30 ^a^	5.02 ^a^	29.72 ^b^	41.83 ^c^	3.685	<0.001
	Apparent digestion (g/d)	0.99 ^a^	3.29 ^a^	11.72 ^b^	15.39 ^b^	1.504	<0.001
	Daily excretion (g/d)	12.01 ^c^	21.01 ^c^	66.56 ^b^	98.58 ^a^	7.142	<0.001

Notes: In the same row, values with the same or no letter superscripts indicate no significant difference (*p >* 0.05), while those with different small letter superscripts indicate a significant difference (*p <* 0.05).

**Table 6 animals-12-01770-t006:** Nutrient contents of feces at different sample times.

Item (%)	Groups				SEM	*p*-Value
	D7	D21	D35	D49		
Fat	1.87 ^a^	1.26 ^b^	0.98 ^b^	0.96 ^b^	0.121	0.016
Dry matter	69.53 ^a^	74.60 ^a^	60.50 ^b^	53.53 ^b^	2.102	<0.001
Protein	33.80 ^a^	26.30 ^a^	14.35 ^b^	13.93 ^b^	2.132	<0.001
Acid detergent fiber	4.57 ^a^	8.66 ^a^	13.63 ^b^	11.86 ^b^	0.877	<0.001
Neutral detergent fiber	9.21 ^a^	14.54 ^b^	21.00 ^c^	18.06 ^bc^	1.161	<0.001
Starch	4.05	4.05	4.38	3.15	0.499	0.857

Notes: In the same row, values with the same or no letter superscripts indicate no significant difference (*p >* 0.05), while those with different small letter superscripts indicate a significant difference (*p <* 0.05).

## Data Availability

The sequence files determined in the present study were deposited at the Sequence Read Archive (SRA; http://www.ncbi.nlm.nih.gov/subs/ (accessed on 10 May 2022); SRA accession number: PRJNA836702).

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
