# Peer review of "Periodical Changes of Feces Microbiota and Its Relationship with Nutrient Digestibility in Early Lambs"

_animals, 2022, doi:10.3390/ani12141770_

Round 1

Reviewer 1 Report

Interesting study but I do not find a rationale why to investigate the faecal microbiota. The number of animals (n=6 is very low). Due to the low number of animals the manuscript should be changed to short communication

The Introduction gives general remarks on specific actions of bacteria in the rumen and in the intestine, however, in this study the faecal microbiota is analysed. How is the faecal microbiota related to the rumen or intestinal microbiota. Most of the rumen microbes will be digested, so maybe only the lower digestible microbes can be measured in the faeces. On the other hand fungi and protozoa belong as well  to ruminant microflora, are these included in this microbiota analysis.

For example authors state in line fecal nutrient content affected the the abundance of bacteria….the faecal nutrient content does not affect anything, it was the nutrient content at the end of the ileum which was available for fermentation who possibly changed microbiota but that is not measured.

Please include growth data for animal and daily gain,

The digestibilty results does not seem to be correct. First of all, for the nutrients for which digestibilities are given, the amount of that nutrient in the respective feed should be given otherwise it is not possible to recalculate your digestibilty data, so please include for starter diet  and milk replacer (Table 1) lipid content, ADF and starch content. These data should be available as you calculated a daily intake of the nutrients, which can only be done when you have the content in the diet and amount fed per day .

Table 4 The daily Intake of NDF and ADF does not make any sense. First of all NDF content is for sure higher than ADF content in the starter diet, but strangely ADF intake is higher than NDF intake and digestibility for NDF is lower compared ADF ….this is impossible!  In addition, in Table 1 NDF content is given as 0.0 but D7 group has received 1 g of NDF and 2about 3 g of ADF???

Please clearly give the feed intake of each feed (milk replacer and staretr ) for each experimental period.

In Table 2 four different indices are shown for alpha diversity but there is no explanation what they measuring, do they all measure the same? PLease improve.

Line210 simpson index showed bidirectional dynamics?? IT is not significant, why should it show any dynamics?

Table 3  PLease indicate which unit these values have

Table 4 has no title

The discussion often repeats results but do not explain the results in any mechanistic way, furthermore due to the wrong digestibilities the discussion is possibly incorrect.

Conclusion is just a repetition of results. The conclusion should indicate the relevance of the results for lambs gastrointestinal tract development. Do data give any hints to improve development or functionality of gastrointestinal tract. WHat is the impact on nutrition of lambs at this age.,  

Author Response

Point 1 Interesting study but I do not find a rationale why to investigate the faecal microbiota. The number of animals (n=6 is very low). Due to the low number of animals the manuscript should be changed to short communication.

Author Response: Thank you for your interest and thoughtful comments on the manuscript. First of all, each region of the gastrointestinal is spatially specialized depending on factors including physiology substrate availabilities, retention time of digesta, and pH levels [1]. These factors are all expected to have a profound impact on the local microbial assemblages and functions, thereby affecting the digestive, immunological, metabolic, and endocrinological processes in ruminants [2]. The rectal fecal flora of ruminants is highly similar to that of the colon and cecum [3], and fecal samples are easy to collect. Increased understanding of the relationship between fecal microorganisms and nutrient utilization in ruminants has a positive effect on reducing feed cost and improving animal productivity. We have further clarified the research background and purpose in the introduction according to your comments.

Secondly, as you mentioned that the number of animals is too low. The main reason is that during the 49-day feeding period, continuous stool sample collection and digestion tests are required, which makes it difficult to ensure the accuracy of sampling of each individual with an increased number of animals. In this study, comparison of the same individuals at different ages can reduce the error caused by individual differences. The data were analyzed by one-way ANOVA with 5 degrees of freedom, the number of samples in each group could meet the statistical requirements.

[1] Stevens, C.E.; Hume, I.D. Contributions of microbes in vertebrate gastrointestinal tract to production and conservation of nutrients. Physiol Rev 1998, 78, 393-427, doi:10.1152/physrev.1998.78.2.393.

[2] Martinez-Guryn, K.; Leone, V.; Chang, E.B. Regional Diversity of the Gastrointestinal Microbiome. Cell Host Microbe 2019, 26, 314-324, doi:10.1016/j.chom.2019.08.011.

[3] Xie, F.; Jin, W.; Si, H.; Yuan, Y.; Tao, Y.; Liu, J.; Wang, X.; Yang, C.; Li, Q.; Yan, X.; et al. An integrated gene catalog and over 10,000 metagenome-assembled genomes from the gastrointestinal microbiome of ruminants. Microbiome 2021, 9, 137, doi:10.1186/s40168-021-01078-x.

Point 2 The Introduction gives general remarks on specific actions of bacteria in the rumen and in the intestine, however, in this study the faecal microbiota is analysed. How is the faecal microbiota related to the rumen or intestinal microbiota. Most of the rumen microbes will be digested, so maybe only the lower digestible microbes can be measured in the faeces. On the other hand, fungi and protozoa belong as well to ruminant microflora, are these included in this microbiota analysis.

Author Response: Thank you very much for your comments, which are very enlightening to our research. We have revised the introduction based on your comments. Researchers have been mainly focused on understanding the rumen microbiota's contribution to the host in the last decade. However, as stated in the response to Point 1, each region of the gastrointestinal is spatially specialized depending on factors including physiology substrate availabilities, retention time of digesta, and pH levels. These factors are all expected to have a profound impact on the local microbial assemblages and functions. The rectal fecal flora of ruminants is highly similar to that of the colon and cecum, and fecal samples are easy to collect. Increased understanding of the relationship between fecal microorganisms and nutrient utilization in ruminants has a positive effect on reducing feed cost and improving animal productivity.

Fungi and protozoa do have a certain effect on ruminant digestive physiology. However, in the procedure of 16S rDNA high-throughput sequencing, fungi and protozoa need to be specially studied by using different universal primers, and this study, like most studies, is mainly focused on bacteria.

Point 3 For example authors state in line fecal nutrient content affected the abundance of bacteria. the faecal nutrient content does not affect anything, it was the nutrient content at the end of the ileum which was available for fermentation who possibly changed microbiota but that is not measured.

Author Response: Each region of the gastrointestinal is spatially specialized depending on factors including physiology substrate availabilities, retention time of digesta, and pH levels [1]. A sheep's colon is 4 to 5 meters long, where the digesta resides for a long time. Microbes multiply rapidly under favorable conditions. It has been reported that bacteria divide every 20 minutes. Therefore, we believe that nutrient content of rectal feces is an important factor affecting fecal flora. Indeed, it is of great significance to study the effects of nutrient content at the end of ileum on microorganisms, but ileum samples of the same sheep at different ages could not be collected in this experiment.

  1. Stevens, C.E.; Hume, I.D. Contributions of microbes in vertebrate gastrointestinal tract to production and conservation of nutrients. Physiological Reviews 1998, 78, 393-427.

Point 4 Please include growth data for animal and daily gain.

Item

Groups

SEM

P-value

D7

D21

D35

D49

Milk replacer intake (kg/d)

0.63

0.70

/

/

/

/

Starter intake (kg/d)

0.02b

0.06b

0.39a

0.49a

0.033

<0.001

Body weight (kg)

4.63c

5.72c

7.74b

10.17a

0.572

<0.001

Average daily gain (kg/d)

0.20a

0.08b

0.14c

0.17cd

0.024

<0.001

Author Response: Thank you for your comments, and we have added the data of milk replacer intake, starter intake, body weight and average daily gain in table 4 according to your suggestion.

Point 5 The digestibilty results does not seem to be correct. First of all, for the nutrients for which digestibilities are given, the amount of that nutrient in the respective feed should be given otherwise it is not possible to recalculate your digestibilty data, so please include for starter diet and milk replacer (Table 1) lipid content, ADF and starch content. These data should be available as you calculated a daily intake of the nutrients, which can only be done when you have the content in the diet and amount fed per day.

Author Response: Thank you for your comments, and we have added starch, fat and ADF contents in table 1 according to your suggestion.

Point 6 Table 4 The daily Intake of NDF and ADF does not make any sense. First of all, NDF content is for sure higher than ADF content in the starter diet, but strangely ADF intake is higher than NDF intake and digestibility for NDF is lower compared ADF this is impossible! In addition, in Table 1 NDF content is given as 0.0 but D7 group has received 1 g of NDF and 2about 3 g of ADF???

Author Response: Thank you so much for your carefully check. We are very sorry for our negligence. When compiling Table 5, we reversed the NDF and ADF in the table header. We have checked the data, revised the table and the relevant presentation.

In Table 1, the NDF content of milk replacer was 0, but the NDF content of starter was 18.87%. The nutrient intake of group D7 was calculated based on the daily intake at 7-10 days of age, including both milk replacer and starter feed intake.

Point 7 Please clearly give the feed intake of each feed (milk replacer and starter) for each experimental period.

Author Response: Thank you for your comments, and we have added the data of milk replacer intake and starter intake for each experimental period according to your suggestion in table 4.

Item

Groups

SEM

P-value

D7

D21

D35

D49

Milk replacer intake (kg/d)

0.63

0.70

/

/

/

/

Starter intake (kg/d)

0.02b

0.06b

0.39a

0.49a

0.033

<0.001

Body weight (kg)

4.63c

5.72c

7.74b

10.17a

0.572

<0.001

Average daily gain (kg/d)

0.20a

0.08b

0.14c

0.17cd

0.024

<0.001

Point 8 In Table 2 four different indices are shown for alpha diversity but there is no explanation what they measuring, do they all measure the same? Please improve.

Author Response: Thank you for your comments, and we have added the meaning of each alpha diversity indices in the discussion according to your suggestion as follows:

Among these diversity indices, the Shannon index measures uncertainty about the identity of species in the sample, and its units quantify information, while the Simpson index measures a probability, specifically, the probability that two individuals, drawn randomly from the sample, will be of different species [4]. Coverage is that the proportion of individuals in the community belonging to undetected species can be estimated reliably, based only on the frequencies of species already in the sample [5]. Chao1 was asymptotic richness estimators and could predict the community diversity [6].

[4] Roswell, M.; Dushoff, J.; Winfree, R. A conceptual guide to measuring species diversity. Oikos 2021.

[5] Chao, A.; Kubota, Y.; Zelen, D.; Chiu, C.; Colwell, R.K. Quantifying sample completeness and comparing diversities among assemblages. Ecological Research 2020, 35, 292-314.

[6] Haegeman, B.; Hamelin, J.; Moriarty, J.; Neal, P.; Dushoff, J.; Weitz, J.S. Robust estimation of microbial diversity in theory and in practice. The ISME journal 2013, 7, 1092-1101, doi:10.1038/ismej.2013.10.

Point 9 Line210 simpson index showed bidirectional dynamics?? IT is not significant, why should it show any dynamics?

Author Response: Thank you for your thoughtful comments, we sincerely apologize for our misstatement, and we have checked and rewritten “bidirectional dynamics” with “the Shannon and Simpson indices index increased from 7 to 21 days of age and then start to decrease after weaning”.

Point 10 Table 3 Please indicate which unit these values have.

Author Response: We appreciate and thank for the detailed review of our manuscript, and the unit (%) have been added according to your suggestion.

Point 11 Table 4 has no title.

Author Response: Thank you for pointing out our mistake and we apologize for our careless. We have added the title “Effects of different sampling times on apparent digestibility in early lambs.” in table 5 according to your suggestion.

Point 12 The discussion often repeats results but do not explain the results in any mechanistic way, furthermore due to the wrong digestibilities the discussion is possibly incorrect.

Author Response: Thanks for your comment. We have modified the discussion according to your suggestion. We are very sorry for reversed the NDF and ADF in the table header when compiling Table 4. We have revised the relevant descriptions in the full text.

Point 13 Conclusion is just a repetition of results. The conclusion should indicate the relevance of the results for lambs gastrointestinal tract development. Do data give any hints to improve development or functionality of gastrointestinal tract. What is the impact on nutrition of lambs at this age.

Author Response: Thank you so much for your carefully check, according to your suggestion, we have rewritten the conclusion, as follows:

The intestinal microflora of lambs changed significantly with age, and up to 49 days old was still an important period of microflora development. The apparent digestibility of dry matter, protein, fat, neutral detergent fiber and acid detergent fiber decreased rapidly with the increase of starter intake from 7 to 35 days of age, especially after weaning. Nutrient intake and digestion are major factors that influence the fecal microbiota by affecting the composition of fermentable substrates in feces. The findings expand our understanding of the gut symbiotic microbiota in ruminants and provide new insights for investigating the gut microbiota’s role in host production.

Reviewer 2 Report

Review of Manuscript Animals-1752403

The paper aimed at characterizing the change of feces microbiota of lambs between 7 and 49 days old. I have for the authors the following comments:

Major comments

In the abstract, a short description of feeds and feeding must be given. Feeding is the main factor that influences microbiota in feces. Give some numbers to understand the magnitude in the changes in digestibility coefficients.

At the end of the introduction the objectives must be clearly defined. Those that authors wrote were not really the aims of the study. Moreover, avoid writing at the end what was in the study done (L96-102). Delete! For that M&M chapter is used.

Authors aimed at studying the relationship between nutrients digestion and microflora. However, detailed information about diets and feeding is missing. The chemical composition of the starter feed must be given. Daily total feed intake must be also separated into milk replacer and started, and the description of chemical composition of diets over time must be given to understand how this developed.

There are some confounding interpretations about the relationship between digestibility and microbiota (see minor comments)

Minor comments

L128: Here the best term is “digestibility” and more precisely “apparent total tract digestibility”

L129: Were feces samples pooled within this 3-d periods? Clarify

L271: I am not agreeing on using the term “digestive function”. Digestive function in more general and can imply many different things. Please replace this word by digestibility along the manuscript

L357-359: Here I am not agreeing. You state that “the development of the intestinal microflora … affected the development of digestive function (digestibility)”. This is not true because digestibility is a function of the change in the chemical composition of the diets. Lambs ate less milk (highly digestible) and more starter (less digestible) over the time, and microbes updated to change in nutrients available in the intestinal tract. In conclusion, nutrients affected digestibility and this the microflora, and not vice versa. This aspect must be considered to avoid wrong interpretations. It must be clear what affects what! Please consider this for the whole manuscript

L386: To weaning stress or to change in diet composition. For sure to the last. Rewrite

L391: As above! Adapted to the diets, and not “recovered from weaning stress

L396: Which various nutrients? Specify

L398: It is wrong to state that no changes in digestibility proves a “development in digestive function”. No change = no development. Delete 397-398

L416-418: Indeed, the type of substrate (chemical component) available is what affects the microbiota. Whether bacteria promote nutrient digestion cannot be elucidated from this study, because change in digestibility in this study was a function of the changes in chemical composition and the source of nutrients (from milk or starter) which differs in digestibility. Therefore, delete statement “these bacteria promote nutrients digestion”

L424: This does not prove any “development” in digestive function per se. Change in digestibility was a function of the changes in chemical composition of diets and increase in intake of starter and less milk

Author Response

The paper aimed at characterizing the change of feces microbiota of lambs between 7 and 49 days old. I have for the authors the following comments:

Author Response: Thanks for your precious time to revise our manuscript, and we have made corresponding modifications according to your suggestions and opinions.

Major comments

Point 1 In the abstract, a short description of feeds and feeding must be given. Feeding is the main factor that influences microbiota in feces. Give some numbers to understand the magnitude in the changes in digestibility coefficients.

Author Response: Thanks for your suggestion. We have added it in digestibility coefficients in the abstract.

Point 2 At the end of the introduction the objectives must be clearly defined. Those that authors wrote were not really the aims of the study. Moreover, avoid writing at the end what was in the study done (L96-102). Delete! For that M&M chapter is used.

Author Response: Thank you for your comments to our manuscript. We have deleted L96-102 according to your suggestion.

Point 3 Authors aimed at studying the relationship between nutrients digestion and microflora. However, detailed information about diets and feeding is missing. The chemical composition of the starter feed must be given. Daily total feed intake must be also separated into milk replacer and started, and the description of chemical composition of diets over time must be given to understand how this developed.

Author Response: Thank you for your comments. We have added chemical composition of the starter feed (Fat, Starch and ADF) in table 1, and also added milk replacer and starter intake over time in table 4 according to your suggestion.

Items

Starter1

Milk replacer

Ingredients [%]

Alfalfa meal

18.50

Corn

21.00

Extruded corn

22.30

Bran

6.00

Soybean meal

21.50

Extruded soybean

4.00

Corn gluten meal

5.00

Limestone

0.30

Premix2

1.00

NaCl

0.40

Total

100.00

Chemical composition

DM (%)

90.96

96.91

DE (MJ·kg-1)

13.01

/

CP (%)

19.50

23.22 

Fat (%)

1.33

13.20

Starch (%)

33.10

0.00

NDF (%)

18.87

0.00

ADF (%)

8.60

0.00

Item

Groups

SEM

P-value

D7

D21

D35

D49

Milk replacer intake (kg/d)

0.63

0.70

/

/

/

/

Starter intake (kg/d)

0.02b

0.06b

0.39a

0.49a

0.033

<0.001

Body weight (kg)

4.63c

5.72c

7.74b

10.17a

0.572

<0.001

Average daily gain (kg/d)

0.20a

0.08b

0.14c

0.17cd

0.024

<0.001

Point 4 There are some confounding interpretations about the relationship between digestibility and microbiota (see minor comments).

Author Response: Thank you for your comments. We have made the modification according to your suggestion in minor comments.

Minor comments

Point 5 L128: Here the best term is “digestibility” and more precisely “apparent total tract digestibility”

Author Response: Thank you for your comments, we have replaced “digestion text” with “apparent total tract digestibility” according to your suggestion.

Point 6 L129: Were feces samples pooled within this 3-d periods? Clarify

Author Response: Thank you for your comments. The feces were weighed daily and pooled for each 3-d period. We have rewritten the procedure for determining total digestive apparent digestibility to make it clear.

Point 7 L271: I am not agreeing on using the term “digestive function”. Digestive function in more general and can imply many different things. Please replace this word by digestibility along the manuscript

Author Response: Thank you for your comments, we have replaced “digestive function” with “apparent digestibility” according to your suggestion and reviewer3’s point 5, and it has been modified in the whole paper.

Point 8 L357-359: Here I am not agreeing. You state that “the development of the intestinal microflora … affected the development of digestive function (digestibility)”. This is not true because digestibility is a function of the change in the chemical composition of the diets. Lambs ate less milk (highly digestible) and more starter (less digestible) over the time, and microbes updated to change in nutrients available in the intestinal tract. In conclusion, nutrients affected digestibility and this the microflora, and not vice versa. This aspect must be considered to avoid wrong interpretations. It must be clear what affects what! Please consider this for the whole manuscript.

Author Response: Thank you for your interest and thoughtful comments on the manuscript, and we have made modifications according to your suggestions, the details are as follows:

Therefore, the development of the intestinal microflora of lambs up to 49 days old is a key stage, in which lambs eat less milk (highly digestible) and more starter (less digestible) over the time, and microbes updated to change in nutrients available in the intestinal tract.

The problem was also considered in the whole manuscript.

Point 9 To weaning stress or to change in diet composition. For sure to the last. Rewrite

Author Response: Thank you for your comments in our manuscript, we have based on your suggestions and comments: “To weaning stress or to change in diet composition. For sure to the last.”, Rewrite part discussion: change in diet composition after weaning.

Point 10 As above! Adapted to the diets, and not “recovered from weaning stress”.

Author Response: Thank you for your comments, we have delated “and recovered from weaning stress” in our manuscript according to your suggestion.

Point 11 Which various nutrients? Specify

Author Response: We appreciate for the detailed review of our manuscript. and we have added supplement (protein, starch, fat, DM, ADF and NDF).

Point 12 It is wrong to state that no changes in digestibility proves a “development in digestive function”. No change = no development. Delete 397-398

Author Response: Thank you for your comment, and we have deleted it according to your suggestion.

Point 13 L416-418: Indeed, the type of substrate (chemical component) available is what affects the microbiota. Whether bacteria promote nutrient digestion cannot be elucidated from this study, because change in digestibility in this study was a function of the changes in chemical composition and the source of nutrients (from milk or starter) which differs in digestibility. Therefore, delete statement “these bacteria promote nutrients digestion”

Author Response: Thank you for your thoughtful comments on the manuscript. We agree with your comments and revised this paragraph. Nutrient intake and digestion are major factors influencing the microbiota by influencing the types of substrates available in the digestive tract. However, it is also possible for some bacteria to interact directly with the host and affect nutrient digestion, although this cannot be elucidated from this study. In the revised discussion, we put forward this possibility as a research prospect.

Point 14 L424: This does not prove any “development” in digestive function per se. Change in digestibility was a function of the chassnges in chemical composition of diets and increase in intake of starter and less milk

Author Response: Thanks for your comment. We have deleted it and modified the conclusion according to your suggestion.

Reviewer 3 Report

Authors have developed an interesting study. Some comments:

In M&M section, please include equations and reference used for digestibility and digestion quantification.

In Table 1, chemical composition of starter should include starch, fat and ADF. How to calculate digestion of those components?

Statistical analysis should be clarified. Authors should include model components and ANOVA structure (i.e., degres freedom).

In Table 4 or 5, please include nutrient excretion in feces. Did you quantified urine excretion?

Please review the concepts of digestibility and digestion. May be you measured apparent digestion, and not necessarily digestibility and total digestion.   

Author Response

Authors have developed an interesting study. Some comments:

Author Response: Thank you for your interest in and affirmation of our research, which gives us great confidence to continue to pursue this profession. Thank you again for your precious time to revise our manuscript.

Point 1 In M&M section, please include equations and reference used for digestibility and digestion quantification.

Author Response: Thank you for your comment, and we have added them according to your suggestion. Details are as follows:

The milk replacer, starter and feces were analyzed for DM(drying at 105℃), CP(AOAC International, 2000), EE(AOAC Inter-national, 2000) [22], NDF, and ADF following a previously described method with heat-stable alpha-amylase and sodium sufite used in the NDF procedure[23], and starch using a commercial assay kit (Solarbio, Shanghai, China) according to the manufacturer’s instructions.

Apparent digestibility of protein, starch, fat, DM, ADF and NDF was calculated Eq.:

AD=[(Fi-Ff)/Fi] ×100%

where AD is the apparent digestibility of protein, starch, fat, DM, ADF or NDF (%); Fi is the intake of protein, starch, fat, DM, ADF or NDF (g); and Ff is the fecal output of protein, starch, fat, DM, ADF or NDF (g).

  1. van Heugten, E.; Funderburke, D.W.; Dorton, K.L. Growth performance, nutrient digestibility, and fecal microflora in weanling pigs fed live yeast. Journal of animal science 2003, 81, 1004-1012, doi:10.2527/2003.8141004x.
  2. Van Soest, P.J.; Robertson, J.B.; Lewis, B.A. Methods for dietary fiber, neutral detergent fiber, and nonstarch polysaccharides in relation to animal nutrition. Journal of dairy science 1991, 74, 3583-3597.

Point 2 In Table 1, chemical composition of starter should include starch, fat and ADF. How to calculate digestion of those components?

Author Response: Thank you for your comments, and we have added starch, fat and ADF in table 1 according to your suggestion. In addition, we supplement the determination of DM, CP, EE, NDF, ADF, ASH and starch, and the details are as follows:

Apparent digestibility of protein, starch, fat, DM, ADF and NDF was calculated Eq.:

AD=[(Fi-Ff)/Fi] ×100%

where AD is the apparent digestibility of protein, starch, fat, DM, ADF or NDF (%); Fi is the intake of protein, starch, fat, DM, ADF or NDF (g); and Ff is the fecal output of protein, starch, fat, DM, ADF or NDF (g).

Items

Starter1

Milk replacer

Ingredients [%]

Alfalfa meal

18.50

Corn

21.00

Extruded corn

22.30

Bran

6.00

Soybean meal

21.50

Extruded soybean

4.00

Corn gluten meal

5.00

Limestone

0.30

Premix2

1.00

NaCl

0.40

Total

100.00

Chemical composition

DM (%)

90.96

96.91

DE (MJ·kg-1)

13.01

/

CP (%)

19.50

23.22

Fat (%)

1.33

13.20

Starch (%)

33.10

0.00

NDF (%)

18.87

0.00

ADF (%)

8.60

0.00

Point 3 Statistical analysis should be clarified. Authors should include model components and ANOVA structure (i.e., degres freedom).

Author Response: Thank you for your comment, and we have added them according to your suggestion in the part of Sequence and statistical analysis.

The data of microbial diversity indices, bacterial densities, growth performance, apparent digestibility, daily intake, apparent digestion, daily excretion, and fecal nutrient contents were analyzed using one-way ANOVA and the least significant difference (LSD) post hoc tests in SPSS software (version 25.0; IBM Corp., Armonk, NY, USA) with 5 degrees of freedom. The following statistical model was used: Yij = μ + Ai + eij, where Y is the microbial diversity indices, bacterial densities, growth performance, apparent digestibility, daily intake, apparent digestion, daily excretion or fecal nutrient contents; μ is the mean; A is the age; and e is the residual error.

Point 4 In Table 4 or 5, please include nutrient excretion in feces. Did you quantify urine excretion?

Author Response: Thank you for your thoughtful comments on the manuscript, and we have added nutrient excretion in feces in table 5 according to your suggestion. We did not quantify urine excretion.

Item

Groups

SEM

P-value

D7

D21

D35

D49

Protein

Apparent digestibility (%)

86.53b

83.59b

66.50a

63.09a

2.630

< 0.001

Daily intake (g/d)

28.03a

39.40a

67.39b

94.84c

5.926

< 0.001

Apparent digestion (g/d)

24.26a

32.81ab

45.53b

59.81c

3.464

< 0.001

Daily excretion (g/d)

3.72c

6.58c

21.86b

35.04a

3.574

< 0.001

Starch

Apparent digestibility (%)

84.99

88.65

86.95

87.33

0.695

0.329

Daily intake (g/d)

5.01a

22.40a

114.37b

160.96c

14.038

< 0.001

Apparent digestion (g/d)

4.41a

19.91a

99.51b

140.82c

12.305

< 0.001

Daily excretion (g/d)

0.60c

2.49c

14.86b

20.14a

1.946

< 0.001

Fat

Apparent digestibility (%)

91.09b

90.66b

69.57a

66.87a

2.558

< 0.001

Daily intake (g/d)

14.46c

16.70d

4.47a

6.49b

1.088

< 0.001

Apparent digestion (g/d)

13.17c

15.14d

3.13a

4.37b

1.108

< 0.001

Daily excretion (g/d)

1.29b

1.56b

1.34b

2.13a

0.186

< 0.001

DM

Apparent digestibility (%)

89.40b

86.27b

69.94a

68.78a

2.044

< 0.001

Daily intake (g/d)

118.87a

171.67a

324.36b

456.50c

29.997

< 0.001

Apparent digestion (g/d)

106.22a

147.58a

228.50b

315.14c

19.187

< 0.001

Daily excretion (g/d)

12.65c

24.09c

95.86b

141.35a

9.770

< 0.001

NDF

Apparent digestibility (%)

67.44b

73.93b

54.37a

53.03a

2.702

0.005

Daily intake (g/d)

2.87a

11.06a

65.44b

92.09c

8.113

< 0.001

Apparent digestion (g/d)

2.23a

7.97a

36.13b

49.32c

4.463

< 0.001

Daily excretion (g/d)

0.64c

3.09c

29.30b

42.77a

3.262

< 0.001

ADF

Apparent digestibility (%)

69.45b

68.61b

38.03a

35.90a

4.165

< 0.001

Daily intake (g/d)

1.30a

5.02a

29.72b

41.83c

3.685

< 0.001

Apparent digestion (g/d)

0.99a

3.29a

11.72b

15.39b

1.504

< 0.001

Daily excretion (g/d)

12.01c

21.01c

66.56b

98.58a

7.142

< 0.001

Point 5 Please review the concepts of digestibility and digestion. May be you measured apparent digestion, and not necessarily digestibility and total digestion.

Author Response: We appreciate and thank for the detailed review of our manuscript, we have replaced “digestive function” with “apparent digestibility” in the full text according to your suggestion and reviewers2’s point 5 and point 7.

Round 2

Reviewer 1 Report

The authors have imroved the manuscript to an acceptable  level.

please move line 414 -420 to the statistic section of material and method where it belongs.

Author Response

Point 1 The authors have improved the manuscript to an acceptable level.

Author Response: First of all, thank you very much for all your constructive suggestions and comments on our manuscript, there is no doubt that the improvement of manuscript quality cannot be achieved without your help. Secondly, thank you for your interest and affirmation in this research, which gives us a lot of confidence to continue to engage in this profession. Thank you again for your precious time to revise our manuscript.

Point 2 Please move line 414-420 to the statistic section of material and method where it belongs.

Author Response: Thank you for your thoughtful comments on the manuscript. We have moved line 414-420 to the position of statistic section of material and method according to your suggestion.
